# Silicon Photonic Polarization Multiplexing Sensor with Both Large Range and High Resolution

**DOI:** 10.3390/s20205870

**Published:** 2020-10-16

**Authors:** Shaojie Yin, Xiaoyan Wang, Zhibin Wang, Sanshui Xiao, Xiaowei Guan

**Affiliations:** 1School of Electrical Engineering, Yanshan University, Qinhuangdao 066004, China; yinshaojie@stumail.ysu.edu.cn (S.Y.); ioe@ysu.edu.cn (Z.W.); 2Institute for Future, Qingdao University, Qingdao 266071, China; xiaoyanwang@qdu.edu.cn; 3DTU Fotonik, Department of Photonics Engineering, Technical University of Denmark, Ørsteds Plads, Building 345A, 2800 Kgs. Lyngby, Denmark; saxi@fotonik.dtu.dk

**Keywords:** polarization multiplexing, silicon photonic sensor, Fabry–Pérot resonator, microring resonator, large range, high resolution

## Abstract

A silicon photonic polarization multiplexing (PM) sensor featuring both a large range and a high resolution is proposed and experimentally demonstrated. The sensor includes a Fabry–Pérot (FP) resonator and a microring resonator (MRR) functioning as the sensing parts. With PM technology, the FP resonator only works on the transverse-electric mode while the MRR only on the transverse-magnetic mode. Thus, the proposed sensor can simultaneously achieve a large range with a short FP resonator and a high resolution with a high-Q MRR. Measured results show a range of 113 °C and a resolution of 0.06 °C for temperature sensing, and a range of 0.58 RIU (refractive index unit) with the resolution of 0.002 RIU for analyte refractive index sensing.

## 1. Introduction

Silicon photonic sensors have been intensively investigated for environment monitoring [1,2,3,4,5,6], bio/chemical parameter acquisition [7,8,9,10,11,12,13], etc. For these applications, sensors are usually required to have both a large range and a high resolution, which is unfortunately not a trivial matter. Principally, a microring resonator (MRR) sensor with a large quality factor (Q) can achieve a high resolution like in the case of using low-loss silicon nitride MRRs [14]. However, such MRRs usually possess large footprints (e.g., R = 100 µm [14]), thus limiting sensing ranges due to the small free-spectral range (FSR). Whereas a submicron silicon disk-based sensor is expected to have a large range, the low Q will dramatically restrain its resolution capability [15]. Though ultra-compact high-Q silicon microdisks have been demonstrated [16,17], the interrogation of the sensor based on such disks may suffer from resonance splitting. Optimal fabrication processes have produced very compact silicon MRRs with very high Q (e.g., 2–3 × 10^5^ at a radius of 5 μm [18] and 9.2 × 10^5^ at a radius of 9 μm [19]), but their performances acting as sensors are still unknown. A possible solution is to use the Mach–Zehnder interferometer-coupled MRR to suppress most of the resonances but leave a prominent one [20]. Nevertheless, such a solution may have the drawback of limited fabrication tolerance since coupling ratios at each resonance should be delicately designed. Another possible solution is to utilize the Vernier effect of two cascaded MRRs, which, however, also requires a delicate design of the dispersions of the two MRRs and thus may challenge the fabrication tolerance [21].

Among many other advantages of the silicon photonic on-chip devices, easy handling of the polarization has enabled many functional devices, including the polarization beam splitter (PBS) for optical communications [22,23] or quantum information processing [24,25]. However, there are few results reported on the silicon photonic sensor assisted by the polarization multiplexing (PM) technology. Silicon MRRs operating on both transverse-electric (TE) and transverse-magnetic (TM) polarizations were demonstrated to simultaneously detect the temperature and refractive index [26], or simultaneously detect the temperature and humidity [27]. Here, we demonstrate a silicon photonic sensor assisted by PBSs, with which the TE and TM modes can separately resonate in different types of resonators, that is, the Fabry–Pérot (FP) resonator with a short length only works on the TE mode and is responsible for a large range, while the MRR with a large Q only works on the TM mode and is in charge of a high resolution.

## 2. Structure and Principle

The structure of the proposed PM sensor is illustrated in Figure 1. Starting from the input point *a*, light first propagates into two in-line PBSs consisting of subwavelength grating waveguides (SWGs). The SWG has been formerly designed as a TM-pass polarizer by reflecting TE mode and allowing TM mode to pass through [28]; hence, it indeed functions as an in-line PBS. For the TM mode, light can pass through the two in-line PBSs and arrive at point *b* (see Figure 1) without any resonating. However, for the TE mode, the in-line PBS performs as a Bragg reflector, and two of them form a FP resonator with a length of *L*. Consequently, light with the TE mode arrives at the point *b* with the spectrum modulated by the FP resonator. Located between point *b* and the output point *c* is a multimode waveguide-based PBS. The principle of such a PBS has been explained in other literatures [29,30]. In short, the PBS consists of two single-mode waveguides and a multimode waveguide sandwiched amid them. The widths of these waveguides are chosen to fulfill the phase-matching condition for the TM fundamental mode (TM_0_) in the single-mode waveguide and the first-order TM mode (TM_1_) in the multimode waveguide. The TE mode in the single-mode waveguide is far from the phase-matching condition with any modes in the multimode waveguide. Consequently, only the TM mode can be coupled into the multimode waveguide and then the MRR connected to the PBS. Therefore, when arriving at *c*, TE mode exhibits the spectrum only modulated by the FP resonator, while the TM mode is only modulated by the MRR. A short FP resonator is chosen for its large sensing range and a large low-loss MRR with a high Q for high resolution. The inserted table in Figure 1 shows the spectra of the TE and TM modes at different points.

Figure 2a shows the scanning electron microscopy (SEM) image of the fabricated in-line PBS with the SWG. Here, it and the whole sensor as well were fabricated on a silicon-on-insulator (SOI) wafer with 205 nm silicon on top of a 3 μm silica layer. Electron-beam lithography with a ZEP520A resist mask was utilized to pattern the chip, and a reactive-ion etching (RIE) process was used to transfer the patterns to the silicon layer. Inverse tapers were used to couple light between the silicon waveguides and the tapered lensed fibers. A finite-difference method (FDM, Ansys/Lumerical) for analyzing the modes of a waveguide and a three-dimensional finite-difference time-domain (3D FDTD, Ansys/Lumerical) method for simulating the light propagation in the waveguide devices were employed. In Figure 2a, the widths of the input waveguide, the narrower end of the taper for connecting the SWG and the input waveguide, and the bridge inside the SWG are 500, 225 and 150 nm, respectively. The pitch of the grating is 440 nm with a 50:50 duty cycle. Figure 2b shows the simulated and measured spectra of the fabricated in-line PBS with the grating period number of 25. The transmission of the TE mode is obtained by connecting a circulator to the input fiber and measuring the reflectance. Here, the simulated and measured spectra are all normalized to the transmission of a straight silicon waveguide. Generally, the measured spectra agree well with the simulated ones, especially for the TM mode. The more pronounced noise in the measured spectrum and the larger discrepancy between the measurement and the simulation for the TE mode may be ascribed to the circulator. Nevertheless, the measurements show that the fabricated in-line PBS performs well in a broad wavelength range, that is, >72% of TE-polarized light being reflected and >90% of TM-polarized light passing through from 1480 to 1580 nm. The beam-splitting function of the in-line PBS can also be clearly seen by the simulated light propagation of the TM mode and the TE mode at 1550 nm (see Figure 2c,d), respectively.

The fabricated multimode waveguide-based PBS is shown in Figure 3a. Here, the width of the multimode waveguide is designed to be 1300 nm to fulfill the phase-matching condition for the TM_0_ mode in the 500-nm-wide single-mode waveguide and the TM_1_ mode in the multimode waveguide. The gap between the single-mode bus waveguide and the multimode waveguide is 200 nm. Figure 3b,c shows the simulated and measured spectra for the PBS when it operates on the TM and TE modes, respectively. Here, the coupling length is chosen as 10.74 μm to allow for a complete coupling period, and we do not take the MRR into consideration. The measured results agree well with the simulation, showing that TM-polarized light can be completely coupled back and forth from/into the multimode waveguide, and that TE-polarized light is hardly coupled into the multimode waveguide. Figure 3d,e further presents the splitting function of the PBS with the aid of full-wave simulation for the TM and TE modes at 1550 nm, respectively.

For the proposed silicon photonic PM sensor, we have also designed the length of the FP resonator *L*, radius of the MRR *R* and the distance of the MRR to the multimode waveguide to be 25 μm, 100 μm and 700 nm, respectively. The measured spectra of the fabricated sensor are shown in Figure 4, which are normalized to a straight waveguide. They clearly show the expected functions, that is, FP resonating for the TE mode with a large FSR of 9.352 nm and MRR resonating for the TM mode. The losses of the sensor for the TE and TM modes around 1550 nm are 2.6 dB and 1.7 dB, respectively. The loaded and intrinsic Qs of the MRR for the TM mode are found to be ~40,000 and ~92,000, respectively, corresponding to a bend waveguide loss of ~5.7 dB/cm.

## 3. Sensing Measurement Results and Discussions

The fabricated silicon photonic PM sensor with silica cladding was measured at different temperatures. The temperature was set and stabilized by a feedback system comprising a semiconductor temperature (LM335) and a semiconductor thermoelectric cooler (TEC) beneath the under-test sample. Figure 5a,b shows the spectra for the TE mode when the temperature coarsely changed from 11.5 °C to 122.3 °C and for the TM mode when the temperature delicately changed from 21.10 °C to 24.40 °C with a step of 0.06 °C, respectively. Here, for the spectra of the TE mode in Figure 5a, the measured results are presented by open markers, and the solid lines are the fitting curves to extract the constructively resonating wavelengths. The fitting curves follow the function of *T* = 1/(1 + *F*sin^2^(*δ*/2)), where *T* is the transmission of the FP resonator, *F* is only determined by the reflectance of the subwavelength grating and *δ* is determined by the wavelength and the resonator length. One can find that the construction wavelength still varies in one FSR despite a large temperature range of ~111 °C. From Figure 5b, one can see the spectra at different temperatures are clearly distinguishable with each other, showing efficient detection of the very small temperature variations and suggesting a high resolution. Furthermore, curves (not shown) following the Lorentzian function were also used to fit the spectra to extract the MRR resonant wavelengths. Figure 5c,d shows the resonating wavelengths for the TE and TM modes at different temperatures, respectively. It is worth noting that the coefficient of determination (*R*^2^) of the aforementioned fittings is >90% for all fittings, suggesting reliable extracted wavelengths. Linear fittings in Figure 5c,d give a temperature sensitivity of 79 pm/°C and 37 pm/°C with a good agreement with the simulations, that is, 76 pm/°C and 43 pm/°C for the TE and TM modes, respectively. Considering a minimal wavelength shift equivalent to 1/15 of the resonator linewidth (39 pm) can be easily detected [12], the minimal detectable temperature change (i.e., resolution) is 0.07 °C, agreeing well with our measurements. The resolution can be potentially further improved to, for example, 0.04 °C by enlarging the distance from the MRR to the multimode waveguide and, thus, the loaded Q since the intrinsic Q of the MRR is ~92,000, corresponding to a minimal linewidth of 17 pm.

We have further exploited the fabricated silicon photonic PM sensor (with the same structural parameters as the above one but without the top silica cladding) for detecting the analyte refractive index (RI). Here, we used a series of index-matching fluids with RIs from 1.400 to 1.458 with a step of 0.002 to verify the performances of the sensor. The fluids were dropped on the sensor for measurements and then rinsed off with acetone/isopropanol/de-ionized water and dried by a nitrogen gun for the next round of measurements. Figure 6a,b shows the measured spectra of the fabricated sensor cladded with the fluids with different RIs for the TE and TM modes, respectively. For the TE mode, the measurement results are shown with open markers, and fitting curves are shown with the dashed lines, while for the TM mode, only the measurement results are shown. Figure 6c shows the measured resonant wavelength shifting with the cladding RI for both TE and TM modes as well as the linear fittings (dashed lines), which give sensitivities of 277 nm/RIU and 97 nm/RIU for the TE and TM modes, respectively. Here, RIU represents the refractive index unit. Although the measured RI step here is 0.002, the highest capable resolution is 2 × 10^−5^ RIU for the present sensor, considering a MRR linewidth of 86 pm and a sensitivity of 277 nm/RIU for the TM mode. Moreover, the attainable RI range is 0.096 RIU, considering a FP FSR of 9.352 nm and a sensitivity of 97 nm/RIU for the TE mode, larger than the measured 0.058 RIU here.

## 4. Conclusions

In summary, we have proposed and experimentally demonstrated a silicon photonic sensor with both a large range and a high resolution by taking advantage of polarization multiplexing technology. With PM technology, the proposed sensor can function as different resonators at different polarizations, that is, FP resonator for the TE mode and MRR for the TM mode, to simultaneously achieve a large sensing range and a high resolution. Measurements have shown a temperature sensing range of 111 °C with a resolution of 0.06 °C, and a refractive index sensing range of 0.058 RIU with a resolution of 0.002 RIU. These achievements are found to be comparable to or better than the state-of-the-art results. For example, the cascaded MRR-based temperature sensor exhibited a resolution of 0.18 °C and a range of 56.85 °C, which can be potentially enhanced to be 568.5 °C [21]. Meanwhile, the potential detectable range and resolution are even better than the measurement results here. Moreover, the sensor performances can be further improved, for example, for an even higher resolution by increasing the Q of the MRR. However, one should note that the wavelength noise will impose limitations on the resolution that can be practically reached in a MRR sensor [31,32]. Such a polarization multiplexer sensor can work more feasibly if combined with a tunable on-chip polarization rotator [33] or polarization controller [34]. The concept of incorporating polarization multiplexing technology into optical sensors and successful experimental validation in this work open new windows for on-chip silicon photonic sensors with versatile or extended functions.

## Figures and Tables

**Figure 1 sensors-20-05870-f001:**
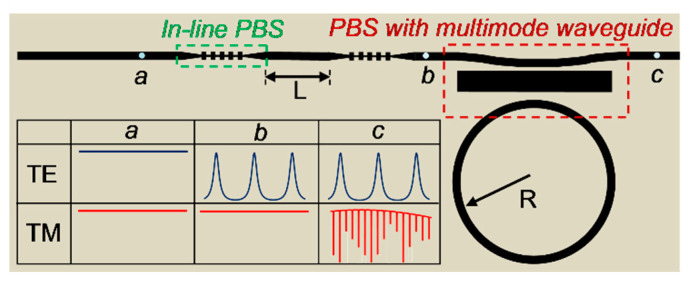
Schematic of the proposed polarization multiplexing sensor. Inserted table shows the spectra of the transverse-electric (TE) and transverse-magnetic (TM) modes at different points in the sensor.

**Figure 2 sensors-20-05870-f002:**
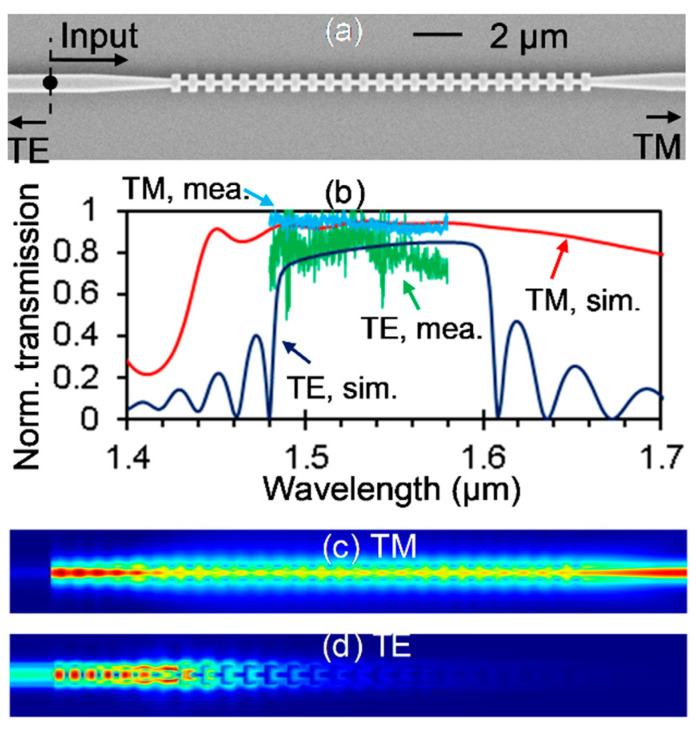
The fabricated in-line polarization beam splitter (PBS). (**a**) SEM image. (**b**) Simulated and measured spectra. Simulated light propagation at 1550 nm for the TM (**c**) and TE (**d**) modes.

**Figure 3 sensors-20-05870-f003:**
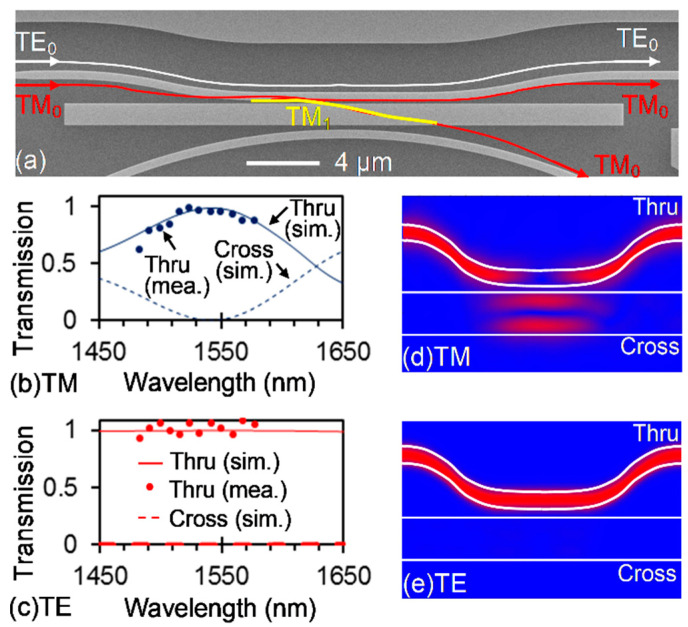
The fabricated multimode waveguide-based PBS. (**a**) SEM image. Simulated and measured spectra for TM (**b**) and TE (**c**) modes when the coupling length is 10.74 μm. Simulated light propagation at 1550 nm when the PBS is operating on TM (**d**) and TE (**e**) modes.

**Figure 4 sensors-20-05870-f004:**
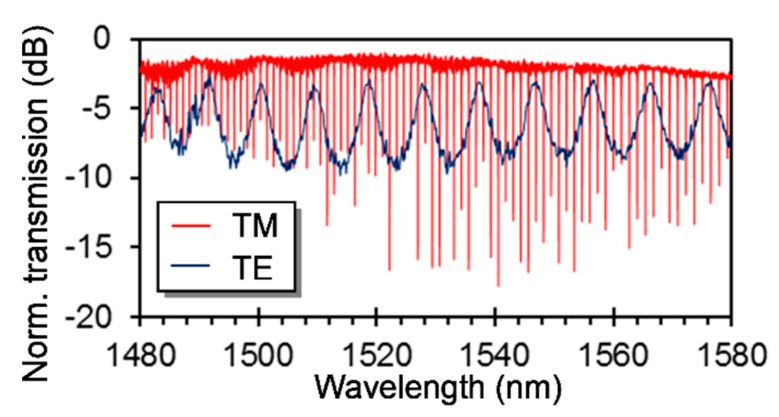
Measured spectra of a fabricated silicon photonic polarization multiplexing sensor.

**Figure 5 sensors-20-05870-f005:**
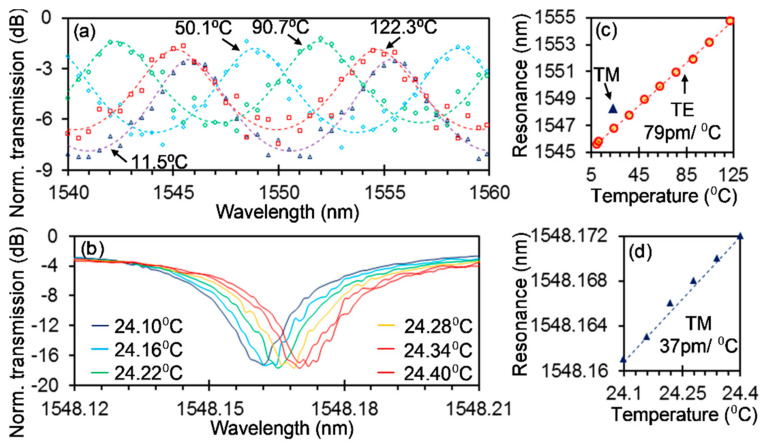
The fabricated silicon photonic polarization multiplexing sensor for temperature sensing. Measured spectra at different temperatures for the TE (**a**) and TM (**b**) modes. For the TE mode, the measurement results are shown with open markers, and the fitting curves are shown with dashed lines, while for the TM mode, only the measurement results are shown. Resonant wavelength with respect to the temperature for the TE (**c**) and TM (**d**) modes. Linear fittings (dashed lines) and slope values are also shown in (**c**,**d**).

**Figure 6 sensors-20-05870-f006:**
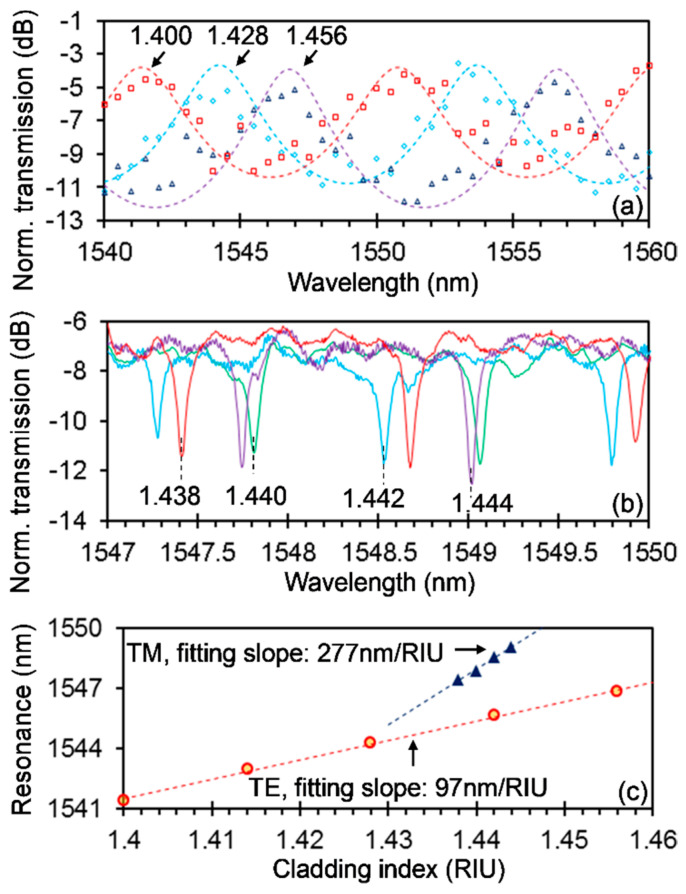
The fabricated silicon photonic polarization multiplexing sensor for refractive index sensing of the cladding analyte. Measured spectra at different cladding refractive indexes (RIs) for the TE (**a**) and TM (**b**) modes. For TE mode, the measurement results are shown with open markers, and fitting curves are shown with dashed lines, while for TM mode, only the measurement results are shown. (**c**) Resonant wavelength with respect to the cladding RI. Linear fittings (dashed lines) and slope values are also shown in (**c**).

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
