# Peer review of "Silicon Photonic Polarization Multiplexing Sensor with Both Large Range and High Resolution"

_sensors, 2020, doi:10.3390/s20205870_

Round 1
Reviewer 1 Report
In this manuscript, the authors propose a solution to the problem of achieving both large dynamic range and resolution with a silicon photonic sensor. In particular, they exploit the Fabry-Pérot fringes generated for TE-polarised light between two in-line polarisation beam splitters as the large free spectral range sensing part – therefore enabling for large dynamic range sensing. The TM-polarised light is instead coupled into a microring resonator further down the propagation direction and employed for the high-Q and high-sensitivity sensing function.
I did enjoy reading the manuscript, which is well written and organised. Explanations and methodology are both clear and sound. The experimental results seem to agree decently well with simulations and compare satisfactorily with the existing literature. Even though the RI and temperature limits of detections are not significantly improved compared to the state of the art, the double functionality relying on polarisation multiplexing used here is interesting and worth exploring, and therefore might be of interest to the photonics and optical sensing community.
Therefore, overall, I recommend the manuscript to be published in Sensors. I only have a few minor concerns listed below, that I believe the authors should address to provide more background literature as well as some further clarifications:
- In the introduction and throughout the manuscript, the authors should cite a few papers that I believe might be of interest to the reader:
- Bogaerts, W., et al. "Silicon microring resonators." Laser & Photonics Reviews 6.1 (2012): 47-73.
- Pitruzzello, G. and Krauss, T. F. "Photonic crystal resonances for sensing and imaging." Journal of Optics 20.7 (2018): 073004.
- Hu, J., et al. "Design guidelines for optical resonator biochemical sensors." JOSA B 26.5 (2009): 1032-1041. (this third one is related to my next point)
- Regarding the measurements taken at different temperatures, I think it would be useful to state how the authors changed the temperature and, perhaps more importantly, how they ensured stability during measurements, especially when they measured the small changes in figure 5(b) with the TM ring modes. In fact, according to Hu et al. (see reference above), the performance of such a high-Q resonance is limited by temperature fluctuations, which might indeed influence the measurements. The authors should comment on this aspect to make sure that the temperature measurements are reliable.
- From figure 2(b) it seems that the transmission of a single PBS is quite noisy. However, the FP fringes in figure 5(a) appear much cleaner, so that the shifts can be easily identified to produce figure 5(c). Can the authors please comment on this?
- How were the peak/dip wavelengths identified, both from the FP fringes and the MRR modes? Were the spectra fitted? The authors should provide this information.
- The reported data points in figure 5(c), (d) and 6(c) should include error bars, which presumably come from the error in determining the peak/dip wavelengths? Were repeats conducted?
Reviewer 2 Report
Some modifications should be introduced in the contents:
1.Grammar corrections and explanations in some phrases such as in page 1 or page 2:
“While ultra-compact high-Q silicon microdisk resonators have been demonstrated [15,16], the microdisk sensor may suffer from the resonance splitting. ….”
“Silicon MRRs were ever demonstrated to work on both transverse-electric (TE) and 43 transverse-magnetic (TM)---“
A careful grammar revision is appreciated.
2.Figures for the inline PBS and multimode waveguide PBS are provided. Text, labels and titles are correct in size and visibility. But references on simulation software used should be included.
3.Discussion on the sensor properties shows the high resolution (TM functioning) and long range (TE functioning) of the sensor: Transmission vs Wavelength and Resonance vs Cladding Index (RIU).
The authors claim that “Furthermore, the potential detectable range and resolution are even better than the measurement results here.” And explanation should be provided. Is it an insight ori s it guided by simulation or experimental results?
4.Although the introduction of polarization multiplexing technique is interesting from the point of view of optical sensors, there is no comparison with other recent experimental results to show its validity. And no comparison with other techniques like cascade microring resonators. A comparison with other techniques and recent results is valuable.
E.g:
- Hyun-Tae Kim and Miao Yu, "Cascaded ring resonator-based temperature sensor with simultaneously enhanced sensitivity and range," Opt. Express 24, 9501-9510 (2016)
5. More theoretical description is needed to provide performance parameters for the dual resonator structure. FSR, resolution,...
Reviewer 3 Report
The authors proposed a silicon photonic polarization multiplexing (PM) sensor which is capable of offering both large range and high resolution. Experimental results were presented to demonstrate the sensor performance over temperature and refractive index change. The paper is well organized and presented. The reviewer think it can be published after minor revisions.
- In the work there are two types of PBSs used. Could it be better to define it with some adjectives to avoid any confusion, e.g., SWG based PBS, MRR based PBS?
- In lines of 82 and 83, it reads “The transmission of the TE mode is obtained by connecting a circulator to the input fiber and measuring the reflectance.” It seems to me what you obtained is the reflectance rather than transmission, and what you plotted in Fig. 2b is the TE reflection. Also, I agree that the measured TM transmission is in good agreement with the simulation, but this is not the case for TE transmission. Could it be that you put the simulated TE reflection together with the measured TE transmission?
- In line 124, it reads “…the measured results are presented by open circles…”. Actually, the open circles in Fig. 5a only represent the measured results for 90.7 degree. You may need to replace “open circles” with “open markers”. The same correction needs to be done for Fig. 6a.
